# General practitioners' decision-making process to prescribe pain medicines for low back pain: a qualitative study

Giovanni E Ferreira [1,2] Joshua Zadro [1,2] Caitlin Jones,[1,2] Julie Ayre [3]
Christine Lin [1,2] Bethan Richards,[1,4] Christopher Needs,[4]
Christina Abdel Shaheed [1,2] Andrew McLachlan,[5] Richard O Day [6]
Christopher Maher[1,2]

For numbered affiliations see end of article.

**Correspondence to**
Dr Giovanni E Ferreira;
giovanni.ferreira@sydney.edu.au

## ABSTRACT

**Background** Pain medicines are widely prescribed by general practitioners (GPs) when managing people with low back pain (LBP), but little is known about what drives decisions to prescribe these medicines.

**Objectives** The aim of this study was to investigate what influences GPs' decision to prescribe pain medicines for LBP.

**Design** Qualitative study with in-depth interviews.

**Setting** Australian primary care.

**Participants** We interviewed 25 GPs practising in Australia experienced in managing LBP (mean (SD) age 53.4 (9.1) years, mean (SD) years of experience: 24.6 (9.3), 36% female). GPs were provided three vignettes describing common LBP presentations (acute exacerbation of chronic LBP, subacute sciatica and chronic LBP) and were asked to think aloud how they would manage the cases described in the vignettes.

**Data analysis** We summarised GP's choices of pain medicines for each vignette using content analysis and used framework analysis to investigate factors that affected GP's decision-making.

**Results** GPs more commonly prescribed opioid analgesics. Anticonvulsants and antidepressants were also commonly prescribed depending on the presentation described in the vignette. GP participants made decisions about what pain medicines to prescribe for LBP largely based on previous experiences, including their own personal experiences of LBP, rather than guidelines. The choice of pain medicine was influenced by a range of clinical factors, more commonly the patient's pathoanatomical diagnosis. While many adhered to principles of judicious use of pain medicines, polypharmacy scenarios were also common. Concerns about drug-seeking behaviour, adverse effects, stigma around opioid analgesics and pressure from regulators also shaped their decision-making process.

**Conclusions** We identified several aspects of decision-making that help explain the current profile of pain medicines prescribed for LBP by GPs. Themes identified by our study could inform future implementation strategies to improve the quality use of medicines for LBP.

## STRENGTHS AND LIMITATIONS OF THIS STUDY

⇒ We used clinical vignettes resembling common low back pain cases to provide clinicians with a concrete starting point where they could base their decisions on.

⇒ Voices and perspectives of younger general practitioners (GPs) or GPs working in rural and remote areas may have been underrepresented.

## INTRODUCTION

Low back pain (LBP) affects over 4 million Australians every year and pain medicines are widely used in its management.[1] Australian data shows that 2 in every 3 patients who seek primary care are prescribed or recommended at least one pain medicine. Examples include over-the-counter analgesics (eg, paracetamol and non-steroidal anti-inflammatories (NSAIDs)) and prescription medicines such as opioid analgesics, gabapentinoids, antidepressants and muscle relaxants.[2]

There is a mismatch between the pharmacological therapies recommended in clinical practice guidelines for LBP and what is prescribed or recommended to patients in primary care. For example, opioid analgesics are one of the most commonly prescribed medicines for people presenting with LBP across various settings and their use has increased 39% in the last decade.[3] This is despite guidelines discouraging their use for LBP when pain is not severe or as first-line care before trialling other treatments.[3 4] Similarly, gabapentinoids are ineffective and potentially harmful for the management of LBP; however their use is increasing.[4–6] On the other hand, the use of pharmacological therapies that are recommended as first-line care such as NSAIDs has reduced 33% from 2004 to 2014.[3] In the USA, antidepressants are prescribed to about a quarter of patients

with LBP.[7] In Portugal, one in seven people with chronic LBP report using antidepressants and benzodiazepines for pain management.[8] However, LBP guidelines recommend against both medicines.[4]

In Australia, LBP is the sixth most commonly managed condition overall and the most common musculoskeletal condition managed by general practitioners (GPs).[9] A UK-based study provided some insight into aspects that shape the decision-making process of health professionals, including GPs. That study described how clinicians select treatments based on knowing the individual patient, the patient's LBP and common treatment options.[10] However, this previous study did not explore the decision-making processes that influence a GP's decision to prescribe pain medicines for LBP in-depth, particularly in relation to the processes that underpin the choice for different pain medicines. Therefore, the aim of this qualitative study was to investigate what influences GPs' decision-making process for prescribing pain medicines for people with LBP.

## METHODS

### Design

This qualitative study was approved by the University of Sydney Human Research Ethics Committee (approval number: 2022/170) and is reported per the consolidated criteria for reporting qualitative research statement.[11]

### Participant selection and recruitment

We recruited 25 GPs registered to practise in Australia who reported having provided care for at least five patients with LBP with or without sciatica in the last 12 months. Four or less is what our team thought would be representative of little to no experience in managing LBP. We defined sciatica as a back problem with radiating pain below the knee with or without neurological symptoms (eg, reduced muscle strength, sensation, or reflexes).[12]

GPs were recruited via a market research company (TKW Research Group, Australia) with a large database of healthcare professionals across Australia and New Zealand. Health professionals opt into this database and therefore have already indicated a willingness to participate in research. None of the participants had a prior relationship with the study investigators. The company sent out emails to GPs in their database and scheduled interviews with those who expressed interest in the study.

### Interviews and data collection

We conducted in-depth, one-on-one semistructured interviews. Interviews were conducted by videoconference between May and June 2022, audiorecorded and transcribed verbatim. A researcher with a background in occupational therapy and experience in qualitative interviews and pain management research conducted the interviews.

An interview guide and three clinical vignettes to guide the interviews were developed by the lead author,

a physiotherapist with experience in LBP research and pain management, and two experienced rheumatologists who provide care for patients with LBP (online supplemental file 1). The vignettes were designed to represent three common LBP presentations: a patient with an acute exacerbation of chronic LBP, a patient with subacute sciatica (pain duration between 6 and 12 weeks) and a patient with chronic non-specific LBP (pain for longer than 12 weeks). Collectively, the vignettes were modelled to describe a range of factors that could play a role in a doctors' decision-making process to manage LBP (table 1). GPs were asked to think aloud how they would manage the cases described in the vignettes.[13] For each clinical decision made by the participant, the researcher facilitating the interview asked participants to elaborate on a point, or to obtain clarification if needed (online supplemental file 1).

### Analysis

Interview data were analysed using two approaches. We used content analysis to create a summary of GP's choices of pain medicines prescribed for each vignette. Content analysis combines both qualitative and quantitative methods, allowing both the content and frequency of categories to be reported.[14 15] Two researchers initially reviewed and familiarised themselves with the transcripts, and coded all pain medicines using the following framework: medicines that would be prescribed as first-, second-, third- or fourth-line care, medicines that GPs would conditionally prescribe (eg, depending on a patient characteristic), and those that GPs were against prescribing.

We used framework analysis to investigate factors that influence GP's decision-making process.[16] Two researchers coded 10 interview transcripts inductively, developed a library of codes in an iterative process, decided on a coding framework (comprising the themes) and applied it to all interviews. Codes, subthemes and themes were reviewed by us, and changes were made as stronger patterns linking codes to themes emerged from the data.[17] Both researchers have a background in physiotherapy and have published on the quality use of pain medicines.[18–20] Data saturation was reached (ie, no new themes emerged).

### Patient and public involvement

It was not appropriate or possible to involve patients or the public in the design, or conduct, or reporting, or dissemination plans of our research.

## RESULTS

We interviewed 25 GPs from April to June 2022. Most were males (64%) practising in metropolitan areas across six states of Australia. Their mean (SD) age was 53.4 (9.1), and they had a mean of 24.5 (9.3) years of practice as GPs. All but three worked full-time in private practices (table 2). The recoding of one interview was mistakenly

**Table 1** Features of each clinical vignette

| | Vignette 1 | Vignette 2 | Vignette 3 |
|---|---|---|---|
| **Patient characteristics** | | | |
| Female | X | | X |
| Male | | X | |
| Middle-aged | X | X | X |
| Physical activity | | | |
| No physical activity | X | | |
| Insufficient physical activity | | | X |
| Physically active | | X | |
| **Social aspects** | | | |
| Children | | | X |
| Primary carer for older person | X | | |
| **Type of low back pain** | | | |
| Atraumatic back pain | X | X | X |
| Back pain initiated after a well-defined event | | X | |
| Back pain with a clear radicular component | | X | |
| Back pain with referred leg pain | | | X |
| **Pain characteristics** | | | |
| Night pain | X | X | |
| Sleep disturbance | X | | |
| Back pain exacerbated by movements/postures | | X | X |
| Acute-on-chronic pain | X | | |
| Sub-acute (pain between 6 and 12 weeks) | | X | |
| Chronic (pain >12 weeks) | | | X |
| Movement restriction | X | X | |
| Limitations to work | | X | |
| **Comorbidities** | | | |
| Obesity | X | | |
| Gastro-oesophageal reflux disease | X | | |
| High cholesterol | X | | |
| Other musculoskeletal conditions | X | | |
| Diabetes | | | X |
| Depression (mild) | | | X |
| **Healthcare use** | | | |
| General practitioner for low back pain | | X | |
| Physiotherapy for low back pain | | | X |
| Acupuncture for low back pain | | | X |
| Psychologist for mild depression | | | X |
| Previous imaging | X | | |
| Previous/current use of pain medicines | X | X | X |
| Paracetamol, NSAIDs | X | | X |
| Opioids | | X | |

NSAID, non-steroidal anti-inflammatory.

| Table 2 | Characteristics of participants (n=25) |
|---|---|
| Age, mean (SD) | 53.4 (9.1) |
| Sex, n (%) | |
| Female | 9 (36%) |
| Male | 16 (64%) |
| State/territory, n (%) | |
| New South Wales | 9 (36%) |
| Queensland | 6 (24%) |
| Victoria | 5 (20%) |
| South Australia | 2 (8%) |
| Western Australia | 2 (8%) |
| Tasmania | 1 (4%) |
| Remoteness, n (%)* | |
| Metropolitan | 22 (88%) |
| Rural/remote | 3 (12%) |
| Patients with low back pain seen per year, n (%) | |
| 5–25 | 2 (8%) |
| 26–50 | 4 (16%) |
| 51–100 | 3 (12%) |
| >100 | 16 (64%) |
| Years of experience, n (SD) (min–max) | 24.6 (9.3) (5–40) |
| Weekly clinical workload (hours), mean (SD) (min–max) | 38 (7.7) (14–50) |

*Defined per the Australian Statistical Geography Standard; GPs working in metropolitan areas are those whose practice is located in a Greater Capital City Statistical Area.
GP, general practitioner.

interrupted just before vignette #3 started and therefore there are only data from 24 GPs for that vignette.

### Content analysis

Paracetamol and NSAIDs were commonly prescribed for all three vignettes. However, opioid analgesics were the pain medicines most prescribed or conditionally prescribed for the acute exacerbation of chronic LBP and sciatica vignettes, and antidepressants for the chronic LBP vignette. Codeine was the most prescribed opioid analgesic for the acute exacerbation vignette, whereas tramadol was the most common choice for the sciatica vignette. Concerns over the safety of opioids, particularly codeine, were commonly reported by GPs across all vignettes. Those concerns were often specific to a type of opioid, as many who expressed concerns over one type of opioid would still prescribe a different opioid analgesic or conditionally prescribe the same opioid. For example, this was observed in the interviews of 11 (44%) GPs in the acute exacerbation vignette (table 3).

Anticonvulsants were the medicine class with the highest number of GPs indicating they would not prescribe for the acute exacerbation vignette: nine (36%) were against prescribing them because the patient did not have signs of neuropathic pain that would justify their use. In contrast, 16 (64%) of GPs recommended an anticonvulsant for the sciatica vignette, mostly pregabalin. GPs prescribed antidepressants for all three vignettes, more commonly for the chronic LBP vignette (n=15, 63%). Of those, most chose duloxetine (n=10, 42%) and/or amitriptyline (n=7, 29%). GPs prescribed antidepressants for different reasons for each vignette: comorbid depression influenced decision-making for the chronic LBP vignette, whereas neuropathic pain was the key reason leading to antidepressants being prescribed for the sciatica vignette.

### Factors influencing GP's decision-making to prescribe pain medicines for LBP

We identified five key themes that influenced GP's decision-making. Themes, subthemes and supporting quotes are presented in table 4.

#### Theme 1: prioritising personal information sources
*Subtheme 1: previous experiences*

Past clinical and personal experiences were recognised by most GPs as the most important source of information to help them make decisions. Some acknowledged that accumulated clinical experience contributed to the formulation of shortcuts that guided decision-making. Meaningful past experiences were often centred around success using medicines with patients, but also personal success when they experienced pain themselves.

GPs considered patients' previous treatment experience an important source of information to assist with their decision-making. Giving patients options and discussing those options with them was often mentioned. However, in those circumstances, patient preferences and values appeared to be the key drivers of decision-making. Discussion about the evidence-base for each option was less prominent.

#### Subtheme 2: guidelines are unhelpful

Many GPs reported that guidelines were not helpful. Guidelines were noted to be particularly unhelpful when patients required more than minimal intervention such as advice and simple analgesics. Recommended interventions such as advice to stay active and education about the benign nature of most LBP were seen as problematic as it could be hard to convince patients in pain of their usefulness. One GP noted that the problem with guidelines was not necessarily their recommendations, but the time required to implement them in a context of GPs pressured for time. Clinical practice guidelines for LBP were not mentioned. Only the *Therapeutic Guidelines*[21] were mentioned by a limited number of GPs.

#### Subtheme 3: clinical training

While the applicability of guidelines was questioned, continuing education activities run by specialists and

**Table 3** Frequency of pain medicines that GPs would prescribe, would conditionally prescribe, were against prescribing or that were not mentioned for each vignette

| | Paracetamol | NSAIDs | Opioids | Antidepressants | Anticonvulsants | Oral steroids | Benzodiazepines |
|---|---|---|---|---|---|---|---|
| Vignette 1 (n=25) | | | | | | | |
| Would prescribe | 13 | 17 | 8 | 6 | 1 | 0 | 0 |
| Would conditionally prescribe | 0 | 3 | 14 | 3 | 0 | 1 | 3 |
| Against prescribing | 0 | 2 | 2 | 6 | 9 | 1 | 1 |
| Not mentioned | 12 | 3 | 1 | 10 | 15 | 23 | 21 |
| Vignette 2 (n=25) | | | | | | | |
| Would prescribe | 8 | 16 | 7 | 6 | 11 | 3 | 0 |
| Would conditionally prescribe | 0 | 0 | 12 | 7 | 5 | 2 | 0 |
| Against prescribing | 0 | 2 | 6 | 3 | 5 | 3 | 1 |
| Not mentioned | 17 | 7 | 0 | 9 | 4 | 17 | 24 |
| Vignette 3 (n=24)* | | | | | | | |
| Would prescribe | 8 | 8 | 8 | 10 | 5 | 0 | 1 |
| Would conditionally prescribe | 1 | 1 | 3 | 5 | 2 | 0 | 0 |
| Against prescribing | 0 | 2 | 4 | 0 | 1 | 2 | 0 |
| Not mentioned | 15 | 13 | 9 | 9 | 16 | 22 | 23 |

*Data from 24 GPs are shown. The interview with one GP discussing vignette 3 was not recorded due to technical issues.
GP, general practitioner; NSAIDs, non-steroidal anti-inflammatory.

opinion leaders were highly valued. By comparison, training programmes were rarely mentioned; only one GP with 5 years of experience mentioned the role of medical school and GP registrar training in addressing the current overuse of opioids for LBP management.

### Theme 2: strong reliance on a biomedical model to guide prescription
#### Subtheme 1: pathoanatomical diagnosis
There was a strong sense among GPs that a pathoanatomical diagnosis was key to inform the choice of pain medicine to be prescribed. This was more pronounced for the chronic LBP vignette where, according to GPs, there was a more uncertain clinical presentation due to the description and duration of symptoms as well as comorbidities. Many GPs reported they would have referred the patient for imaging, and many considered the number and nature of imaging findings a key indicator of severity of the problem, which they linked to the prescription of stronger analgesics.

#### Subtheme 2: sleep patterns
Sleep influenced decision-making across the three vignettes, even though the patient in the chronic

LBP vignette had no night pain or sleep disturbance. Improving sleep was an important treatment goal in the acute exacerbation vignette. For some, improving sleep was seen as the primary goal of treatment, and often led GPs to prescribe an opioid analgesics or antidepressants, or both. Advice on sleep hygiene was not mentioned.

#### Subtheme 3: pain severity
There were opposing views on the role of pain severity as a factor that influenced decision-making. While some were happy to consider stronger analgesics when pain was more severe, for others pain severity would not prompt immediate escalation to stronger analgesics. GPs were more inclined to consider escalating to stronger analgesics when discussing the acute exacerbation vignette and less inclined to do so in the chronic LBP vignette (the average pain intensity was the same in both). For some GPs who would not consider escalating to stronger analgesics, their initial choice of treatment was typically a weak opioid analgesic such as codeine. Those preferring codeine mentioned being more used to the prescribing it than other opioid analgesics.

**Table 4** Themes, subthemes and supporting quotes

| Themes | Subthemes | Supporting quotes |
|---|---|---|
| Prioritising personal information sources | Previous experience | ► As an experienced GP of just over 20 years, you have scripts in your mind that bring you to a sort of automatic pilot approach. (GP #20, male, 21 years of experience)<br>► I'd probably just give her the options of weather she wants some anti-inflammatory, something with codeine, and antidepressant. (GP #3, female, 23 years of experience)<br>► I've had radicular pain myself, and the only thing that fixed me was Lyrica. I actually had little faith in Lyrica before I used it, but nothing else touched my pain. A lot of my approach to radicular pain is based on my own experience. (GP #1, female, 30 years of experience) |
|  | Guidelines are unhelpful | ► Sometimes I find guidelines are not actually practical. If they say 'oh tell the patient they need to keep fit and tell them that pain is not a bad sign and that they should keep active and a little bit of pain is ok', it can be hard to convince patients of that. (GP #1, female, 30 years of experience) |
|  | Clinical training | ► Probably what I've heard at specialist meetings when the specialists have spoken. I went to one last night and the neurosurgeon was basically saying you try and avoid going on the path of long-acting opioids, and you can use a quick acting opioid. (GP #6, female, 35 years of experience).<br>► Part of it is what we what we've learned through GP registrar training and medical school, where there's a strong emphasis on not using opioid painkillers in for the management of back pain (GP #17, male, 5 years of experience) |
| Strong reliance on a biomedical model to guide prescription | Pathoanatomical diagnosis | ► When we do the imaging, it's going to support our treatment more. If you find one of two nerves there affecting him, we can start him on pregabalin 25 mg (GP #23, female, 28 years of experience)<br>► If we found multiple levels of a minor disc bulge or something, I would be happier to give her something stronger, but if the MRI just showed that she had a disc bulge at one level and everything else looks fine, there's no nerve impingement, I don't think that would warrant it. I'd be happy to try amitriptyline than other medications as a first line. (GP #9, female, 25 years of experience). |
|  | Sleep patterns | ► I'd be happy to give her some panadeine (NB: codeine+paracetamol) straight away because it's keeping her awake at night, and that's always a bit of a problem. She wakes up because of the pain … I mean you don't want people waking up with pain, so give her a bit of mild pain relief. (GP #22, male, 39 years of experience) |
|  | Pain severity | ► No because she might still respond to what I've decided to use, it might just be temporary that she's eight or nine (NB: pain out of 10) (GP #4, female, 30 years of experience) |
|  | Comorbidities | ► There are some other antidepressants like Cymbalta which can act as a neuropathic agent as well as an antidepressant. So there are antidepressants which we can use which help patients manage their depression as well as well as their pain. (GP #19, male, 30 years of experience) |
| Social factors as root causes | Pain medicines do not address the root cause of the problem | ► We don't want her having a fall with her mum and we don't want her experiencing side effects that may affect her ability to care for her mother. (GP #7, male, 18 years of experience) |
|  | Psychosocial-based prescribing | ► Since she's getting relief with paracetamol and ibuprofen, I think the next step is to find if there are any associated issues in terms of looking after her mum. Can we get some home help … refer mum to aged care services? I mean she might be able to get NDIS help (NB: government support) as well (GP #11, male, 20 years of experience) |

Continued

**Table 4** Continued

| Themes | Subthemes | Supporting quotes |
|---|---|---|
| Judicious use of pain medicines | Keep it simple or make it complex? | ▶ They get flare ups here and there and I think you don't jump up and down as long as you know exactly the origin of the pain and you know you excluded that there is more to it than just what's presenting at this stage, and I think let's just keep it simple. And you can always go back and increase the medication or order further investigations if things don't subside, but I don't hit them harder on the 1st presentation. (GP #15, male, 30 years of experience)<br>▶ I tend to like to use a smaller dose of a lot of things rather than a bigger dose of one or two things. So I'd write a little plan and it would have a sliding scale so it would start off with the Panadol three times a day plus the Nurofen. That's the foundations of the house, and then I would add into that tramadol sustained release 150 to 100 twice a day, and then I would usually add in amitriptyline 10 to 20 milligrams before bed. (GP #25, female, 30 years of experience) |
| Concerns about prescription | Drug-seeking behaviour and risk of long-term misuse | ▶ It's a chronic, insidious, ongoing bit of depression. And it just typically … It's just super clean. She gets started on endone and then get stuck on endone and that becomes their whole life. And they just slowly over a course of years dose-escalate. And they feel better. Because of the euphoria associated with the oxycodone. Terrible. (GP #16, male, 34 years of clinical experience) |
|  | Side effects of prescription pain medicines | ▶ Opioids induce hyperalgesia, which comes with a lot of opioids such as codeine, which is a poor analgesic. I don't use that either. I basically use 3 analgesics in my practice: tramadol, buprenorphine, and tapentadol. (GP #19, Male, 30 years of experience) |
|  | Stigma around prescription | ▶ It could be done, but it would involve a very lengthy consultation and we as GPs don't have half an hour to 45 minutes to discuss something a little bit more wayward. (GP #7, male, 18 years of experience) |
|  | Pressure from regulators | ▶ The guidelines at the moment for non-opioid medication are clear as far as we can all see. I don't know how much that'll go the wrong way and we'll end up not treating anybody pains eventually. We get sued for not treating anybody pain and the pain and suffering and next thing you know the guidelines will change, you know? But you can't win, can you? (GP #14, male, 40 years of experience)<br>▶ Department of Health want us not to give pain medicine at all for low back pain but sometimes it's not realistic. Sometimes we need to help the patient even though I mean, potentially they can be additive of course, but, realistically, the patient is in so much pain, they can't move. This is cruel. Not giving opioid medicines, at least for a few days. I'm not the type of person that gives everybody strong pain medicines, but I think there is a role in some patients. (GP #18, male, 27 years of experience) |

*Subtheme 4: comorbidities*

Comorbidities described in the vignettes were noted by most and influenced decision-making for some. Losing weight was often mentioned as a treatment goal in all three vignettes and led some GPs to preferring non-pharmacological approaches first or avoiding pharmacological treatments known to cause weight gain (eg, antidepressants), despite obesity only being mentioned in the acute exacerbation vignette. The history of mild depression described in the chronic LBP vignette led many GPs to consider an antidepressant. The absence of depression in the subacute sciatica vignette was an important factor for GPs when deciding between different pharmacological options for neuropathic pain symptoms. Many acknowledged the role of antidepressants (eg, duloxetine, amitriptyline) as potentially effective for symptoms described in the sciatica vignette, but most preferred anticonvulsants as the patient was young, otherwise healthy, physically active, and did not have chronic pain. In contrast, most GPs preferred to prescribe duloxetine for patients with comorbid depression and chronic pain such as the patient described in chronic LBP vignette.

### Theme 3: social factors as root causes
*Subtheme 1: pain medicines do not address the root cause of the problem*

Many GPs reported that prescribing medicines for pain for the acute exacerbation vignette would not be likely to address the root cause of the problem.

Some suggested non-pharmacological treatments such as lifestyle interventions, but most GPs expressed concern about the level of social and family support that the patient had and its potential negative impact on her LBP.

### Subtheme 2: psychosocial-based prescribing

Some GPs considered the patient's social circumstances when deciding whether to prescribe a pain medicine and what to prescribe. In the acute exacerbation vignette, many thought that the patient's social circumstances were important contributors to her pain but from different perspectives. While some emphasised the physical strain related to her carer role as a contributor to the pain, others recognised the psychological consequences of the patient's social circumstances. Those who acknowledged physical factors preferred recommending non-pharmacological options such as ergonomic devices, whereas those who recognised the psychological toll of the patient's carer role more commonly prescribed opioid analgesics or antidepressants. Other GPs were wary about whether side effects from opioid analgesics would hinder the patient's ability to care for her mother due to potential side effects, such as sedation.

### Theme 4: judicious use of pain medicines
### Subtheme 1: keep it simple or make it complex?

There were opposing views about judicious use of pain medicines. Many GPs followed the principles of the WHO analgesic ladder and avoided polypharmacy, particularly for the acute exacerbation vignette, and, to a lesser extent, the sciatica vignette. For GPs following that approach, there was a perception that the clinical course of both vignettes would be favourable and keeping prescription of pain medicines to a minimum, combined with non-pharmacological measures, would be effective. Chronic pain and the history of depression were seen as barriers to improvement in the chronic LBP vignette—and therefore to a perception that the patient with chronic LBP would have been more complex to manage.

Other GPs reported that they preferred prescribing multiple pain medicines in combination, at least in the short-term, and most commonly in the sciatica and chronic LBP vignettes. In the sciatica vignette, this was driven by two factors: the notion that there was a well-established mechanism of injury (as opposed to a degenerative process with no known trigger as in the acute exacerbation vignette), and the perception that the clinical presentation was more serious as the patient had radicular symptoms. Both factors led GPs to consider stronger pain medicines such as opioid analgesics and anticonvulsants. In the chronic LBP vignette, polypharmacy often included the prescription of simple analgesics with antidepressants, most commonly duloxetine, which was driven mainly by the history of mild depression.

### Theme 5: concerns about prescription
### Subtheme 1: drug-seeking behaviour and risk of long-term misuse

A key concern for many GPs was drug-seeking behaviour. GPs used multiple strategies to identify drug-seeking behaviours, such as a review of medical history and intuition. Some were reluctant to prescribe opioid analgesics to patients who were not their regular patients. Some GPs mentioned being able to identify certain stereotypes more likely to be drug seekers. Although opioids were commonly considered by GPs across the three vignettes, most had concerns about their potential for misuse and addiction, particularly in relation to the chronic LBP vignette. Chronic pain and a history of depression were seen as risk factors to long-term misuse and addiction to opioid analgesics.

### Subtheme 2: side effects of prescription pain medicines

Similarly, most concerns about side effects were related to opioid analgesics. Many GPs expressed concerns about the safety profile of specific opioid analgesics, most commonly codeine, and made decisions about which opioid analgesic to use based on their knowledge about the safety profile of opioid analgesics (eg, sedation, constipation, nausea). In contrast, very few GPs expressed concerns about other pain medicines with potential for misuse and abuse, such as pregabalin.

### Subtheme 3: stigma around prescription

Some GPs were aware of the stigma around prescribing pain medicines such as antidepressants. Although frequently mentioned as a treatment option for the sciatica and chronic LBP vignettes, some had concerns that prescribing antidepressants had potential for misinterpretation by patients—for example, that they were implying that the patient's pain was psychological. Lack of time during a regular consultation was seen as important barrier for prescribing antidepressants. There was a perception that longer consultations would be required to explain why the medication was being prescribed to mitigate any concerns from patients.

### Subtheme 4: pressure from regulators

Some GPs were concerned about being pressured by regulators to not prescribe pain medicines such as opioid analgesics. Their concerns included the potential consequences of undertreating pain. No GP mentioned that pressure from regulators had changed their prescribing behaviour.

## DISCUSSION

GPs in this study made decisions about what pain medicines to prescribe for LBP largely based on previous experiences, including their own clinical and personal experience of LBP management, not guidelines. The choice of pain medicine was influenced by a range of clinical factors, more commonly the patient's pathoanatomical diagnosis. While many adhered to principles of pain

medicines prescribing, polypharmacy scenarios were also common. Concerns about drug-seeking behaviour, side effects and stigma around prescription also shaped GPs' decision-making process.

Antidepressants were considered by GPs in our study when they thought depression was linked to the patient's symptoms or as a pain medicine to treat neuropathic pain. Although there is some evidence that antidepressants may be effective for those with chronic pain and comorbid depression and neuropathic pain,[20] the role of antidepressants for chronic LBP is limited and the evidence for sciatica is inconclusive.[18] Some GPs voiced concerns about stigma around prescribing antidepressants for pain that was not observed for any other pain medicine in our study. Stigma around the use of antidepressants is a well-recognised phenomenon in the mental health field, where studies have shown that people using antidepressants for depression have a perception that their condition is more severe and that they are weak or unable to cope with their problems.[22] GPs in our study had the perception that patients may not receive the idea of being prescribed an antidepressant for their pain well.

While our GPs were aware of the harms and potential for misuse and abuse of opioid analgesics, fewer mentioned concerns with other prescription pain medicines such as anticonvulsants, which also have potential for misuse, addiction and death.[23] The apparent lack of awareness about the harms of these medicines appears to also be common among the public.[24] In contrast, public health entities and government regulatory agencies have recognised the concerning potential harms of these medicines and have taken action to reduce them. Such concerns have led regulatory agencies in the UK and Australia to up schedule and to add Boxed Warnings to the Product Information and Consumer Medicine Information for pregabalin, respectively.[23 25]

Time constraints in general practice are linked to poor adherence to guidelines, inappropriate prescription and provision of worse care.[26–28] Furthermore, meeting guideline recommendations in primary care requires an unrealistic amount of time that GPs do not have.[29] In line with these findings, there was a perception among some GPs that the time required to implement guideline recommendations into practice, or to discuss prescription of certain pain medicines (eg, antidepressants), was an important barrier to their prescription. Several studies have pointed out that GPs perceive it as nearly impossible to meet all guideline recommendations as there are too many recommendations and most are complex. For example, implementing and documenting all recommendations from guidelines for common problems seen in general practice would take up to 27 hours per day.[30] LBP guidelines involve a complex series of steps from triage to management and may also take considerable time to be implemented.[31]

Our findings provide insights for future research to improve the quality use of medicines for LBP. For example, GPs in our study rarely made decisions based on guidelines or evidence. Rather, they strongly relied on previous personal experience with pain medications, and interactions with patients, colleagues and opinion leaders. Our observations are in line with previous work that has shown that GPs make decisions mainly based on interaction with others rather than endorsed guidelines—a process known as 'mindlines'.[32]

There are opportunities to develop educational interventions to address some of the factors that influenced decision-making that do not rely on guideline dissemination and consider the resources and sources of information most valued by GPs. One such factor is the inaccurate belief that certain pathoanatomic diagnoses on their own should guide the decision to prescribe certain pain medicines. Pathoanatomic findings are often incidental and have been shown to be mostly unrelated to the development, or severity of symptoms.[33] Interventions could also be developed to increase GPs' awareness of their own biases and heuristics as identified by our study.

The Australian Commission on Safety and Quality in Health Care has recently launched a Clinical Care Standard for LBP to improve care for LBP. One of its core messages is around the quality use of medicines. Specifically, they recommend against anticonvulsants, benzodiazepines and antidepressants, and that clinicians only consider opioid analgesics for carefully selected patients at the lowest dose for the shortest duration possible.[34] Strategies to implement the Clinical Care Standard may need to account for the perception among GPs that following guidelines may lead to patients being undertreated. Our study showed that GPs value opinion leaders as a source of information, which suggests that implementation strategies to improve the quality use of medicines for LBP could involve opinion leaders, an approach that has been shown to improve compliance with evidence-based care.[35]

A strength of our study is the use of vignettes, a widely used methodology for assessing decision-making.[36] Vignettes provided a concrete starting point so that GPs could base their decisions on a detailed patient case, rather than answering abstract questions about attitudes and perceptions. However, the decision-making processes described in our study were anchored to the information provided in the vignettes and may not necessarily be generalisable to all LBP cases or truly reflect objective prescription data.[37] For example, while our study did show that GPs would most commonly prescribe or conditionally prescribe opioids for an acute exacerbation of chronic LBP (vignette 1) or subacute sciatica (vignette 2), which agrees with Australian data on prescription medicines for LBP,[3] our finding that GPs would more commonly prescribe antidepressants for the patient with chronic LBP in vignette 3 differs from existing data. It is unknown whether this difference may be due to the characteristics of the vignette and reflective of practice for patients with the characteristics described in that vignette. Our sample was composed mostly of very experienced GPs practising in metropolitan areas. Their views

may differ from those in earlier stages of their careers. For example, the youngest GP in our sample mentioned the role of University and GP training in teaching GPs about the lack of effectiveness and harms of opioids for LBP. That view was not shared by any other experienced GP, which could indicate recent changes in teaching curricula. GPs practising in rural areas are more likely to prescribe opioid analgesics for LBP.[3] However, our sample lacked GPs practising in those areas, limiting our ability to contrast potential differences in drivers of decision-making between those areas.

## CONCLUSIONS

We identified several aspects of decision-making that help explain the current profile of pain medicines prescribed for LBP in by GPs. Themes identified by our study could inform future implementation strategies to improve the quality use of medicines for LBP.

**Author affiliations**
[1]Institute for Musculoskeletal Health, The University of Sydney and Sydney Local Health District, Sydney, New South Wales, Australia
[2]Sydney Musculoskeletal Health, Faculty of Medicine and Health, The University of Sydney, Sydney, New South Wales, Australia
[3]Sydney Health Literacy Lab, School of Public Health, Faculty of Medicine and Health, The University of Sydney, Sydney, New South Wales, Australia
[4]Department of Rheumatology, Royal Prince Alfred Hospital, Sydney, New South Wales, Australia
[5]Sydney Pharmacy School, Faculty of Medicine and Health, The University of Sydney, Sydney, New South Wales, Australia
[6]St Vincent's Clinical School, Department of Clinical Pharmacology & Toxicology, University of New South Wales, Sydney, New South Wales, Australia

**Contributors** Conceptualisation: GEF, JA, CL, BR, CN, CAS, AM, ROD and CM; Methodology: GEF and JA; Validation and formal analysis: GEF and JZ; Investigation: GEF, JZ and CJ; Resources, data curation and project administration: GEF; Supervision: CM; Funding acquisition: GEF, CL, CAS, BR, AM, ROD and CM; Writing—original draft: GEF; Writing—review and editing: JZ, JA, CJ, CL, BR, CN, CAS, AM, ROD and CM. GEF is the guarantor.

**Funding** This study was partially funded by a seed grant from the Australian New Zealand Musculoskeletal Clinical Trials Network (ANZMUSC). GEF, JZ, JA, CL and CM are supported by National Health and Medical Research Council (NHMRC) fellowships (APP2009808, APP1194105, APP2017278, APP1193939 and APP11094283, respectively). CM is also supported by Centre for Research Excellence grants (APP1134856, APP1171459, APP2006545).

**Competing interests** The Sydney Pharmacy School receives funding from GlaxoSmithKline for a postgraduate scholarship supervised by AM. CM has received research grants from various government and not for profit agencies. Flexeze provided heat wraps at no cost for the SHaPED trial for which CM, BR and CN are investigators. Other authors have no interests to declare.

**Patient and public involvement** Patients and/or the public were not involved in the design, or conduct, or reporting, or dissemination plans of this research.

**Patient consent for publication** Not applicable.

**Ethics approval** The University of Sydney Human Research Ethics Committee approved this research (approval number: 2022/170). Participants gave informed consent to participate in the study before taking part.

**Provenance and peer review** Not commissioned; externally peer reviewed.

**Data availability statement** Data are available upon reasonable request. Data are available upon request.

**ORCID iDs**
Giovanni E Ferreira http://orcid.org/0000-0002-8534-195X
Joshua Zadro http://orcid.org/0000-0001-8981-2125
Julie Ayre http://orcid.org/0000-0002-5279-5189
Christine Lin http://orcid.org/0000-0001-6192-7238
Christina Abdel Shaheed http://orcid.org/0000-0003-1258-5125
Richard O Day http://orcid.org/0000-0002-6045-6937

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
