## [Reviewer comments · BMJ Open]

This paper was submitted to a another journal from BMJ but declined for publication following peer review. The authors addressed the reviewers' comments and submitted the revised paper to BMJ Open. The paper was subsequently accepted for publication at BMJ Open.

ARTICLE DETAILS

TITLE (PROVISIONAL)	General practitioners' decision-making process to prescribe pain medicines for low back pain: a qualitative study
AUTHORS	Ferreira, Giovanni; Zadro, Joshua; Jones, Caitlin; Ayre, Julie; Lin, Christine; Richards, Bethan; Needs, Christopher; Abdel Shaheed, Christina; McLachlan, Andrew; Day, Richard; Maher, Christopher

VERSION 1 – REVIEW

REVIEWER	Di Gangi, Stefania University of Zurich
REVIEW RETURNED	18-May-2023

GENERAL COMMENTS	The manuscript is well written and clear. It reports interesting results even if most of them are not new and known in the literature. The methods part regarding the recruitment of participants should contain more information regarding how the GPs were recruited. As also the authors recognized, there could be a bias due to the fact that most of GPs have long experience in the practice and so young GP are not properly represented. So how this happened? Could not have been possible to involve younger GP? Is there a selection bias or other factors influenced that? Moreover, should also be mentioned in the limitation that 25 is a small sample size. Why GPs with at least 5 patients in the last 12 months were considered? There is a rational for the number 5? Participant characteristics reported in Table 2 should be better described. For instance how Remoteness: Metropolitan/Rural was defined? how much is the range (minimum - maximum) for age, years of experience, workload? In the caption of Table 2 should be also reported that n=25 so that the Table could stand alone. In Table 3 please report also % and not only counts. In Table 3 is reported that in Vignette 3 for one GP was not possible to record the interview for technical issues. The authors should report this in the methods part and better explain what's happened and why was not possible to record it again. In the discussion is not clear what are the novelties of this study and what this study in particular add to the literature, as main of the themes are already known in the literature and as the vignettes
--

	are also used in other studies. It could be interesting to discuss if prescriptions to real patients would differ from prescriptions using the vignettes. Other minor comments: - Abstract line 29 - please correct "describe" with "described". Line 41. Please check the sentence - I think "in" should be removed. line 8 introduction I think that "or" should be deleted. Please check/rephrase the sentence.
--	--

REVIEWER	Ivers, Rowena University of Wollongong
REVIEW RETURNED	28-May-2023

GENERAL COMMENTS	Dear authors, Thank you for this insight into general practitioner views on prescribing for lower back pain. The sample size is adequate for analysis for a qualitative study, and the authors have noted that men were overrepresented, and that rural GPs were underrepresented. The vignettes are realistic - in that they have tried other therapies prior to attending the GP. - these are all common scenarios. Description of the themes emerging was executed well. In the quotes, I can see no mention of heat packs, or referral to physical therapies -if these are not mentioned at all it would be worth mentioning specifically that they were not mentioned. Did any specifically mention keeping active/ moving? If referral to rheumatologist, radiologist for steroid injection and neurosurgeon not mentioned also need to specifically note this (these patients would not necessarily require this). Referral to psychology is written into one of the vignettes (1) and physio into 3. There is one quote where the GP mentioned organising care for elderly mother. Although the focus is on medications these are all important aspects of therapy. While GPs did not mention guidelines for treatment of back pain, most GPs would most likely have undertaken training with NPS on use of opioids and treatment of neuropathic pain -did any mention this? If so, this needs to be mentioned as was a large national program. NPS guidelines on lower back pain in primary care were released in 2011 but new framework was only released late in 2022 so GPs would not have seen this at the time of the study. Many of these GPs prescribed benzodiazepines - did any participants prescribe melatonin or other less addictive medications for insomnia? If so, this needs to be mentioned - also mention if they did (or did not) mention advice on sleep hygiene. In terms of prescribing and Subtheme 1, did any participants mention opiate contracts, practice policies or use of real time prescription monitoring (which at that time would have commenced in Vic and maybe in Qld but perhaps not yet in NSW). Did any mention receiving notification regarding their opiate prescribing practices from the Commonwealth government as this program would have occurred prior to these interviews being undertaken. I subtheme 2 - did any mention any specific side effects, such as constipation, sedation (and need for care with driving), addiction or other side effects - If not, need to also mention this. Subtheme - stigma about using antidepressants - the mention of suggesting a mental health diagnosis is raised - did any GPs mention other flowon effects such as effect on the person's life insurance (increases cost of insurance if existing mental health condition). Did any GPs mention risk to themselves or that there is
--

	a risk of verbal abuse or risk of physical harm with some clients if medications not prescribed (not a major point but definitely occurs). In particular, the paragraph regarding using scans to guide treatment is relevant and discussed valid on this point. The paper would have benefited from a general practitioner input, to interpret the findings so as to understand the context of general practice. Best practice in research involving primary care is to include practitioners on the research team.
--	--

VERSION 1 – AUTHOR RESPONSE

REVIEWER 1

1. The manuscript is well written and clear. It reports interesting results even if most of them are not new and known in the literature.

Response to reviewer: Thank you for your positive assessment of our manuscript.

2. The methods part regarding the recruitment of participants should contain more information regarding how the GPs were recruited.

Response to reviewer: We have added more information about recruitment – see below.

GPs were recruited via a market research company (TKW Research Group, Australia) with a large database of healthcare professionals across Australia and New Zealand. Health professionals opt into this database and therefore have already indicated a willingness to participate in research. None of the participants had a prior relationship with the study investigators. The company sent out emails to GPs in their database and scheduled interviews with those who expressed interest in the study.

3. As also the authors recognized, there could be a bias due to the fact that most of GPs have long experience in the practice and so young GP are not properly represented. So how this happened? Could not have been possible to involve younger GP? Is there a selection bias or other factors influenced that?

Response to reviewer: We agree with the reviewer that this is a limitation of our study which we have acknowledged in the discussion. Unfortunately, only a small number of younger GPs were interested in being interviewed or our study we could not recruit more GPs due to budget constraints.

4. Moreover, should also be mentioned in the limitation that 25 is a small sample size.

Response to reviewer: As highlighted by reviewer 2, we believe our sample size is adequate for the purposes of the study. It is typically accepted that saturation can be achieved with fewer than 25 participants in qualitative studies – for example, 9-17 participants as shown by a recent high-quality review of qualitative studies.¹ We achieved saturation with the included sample size. This information has been added to the manuscript:

Codes, sub-themes, and themes were reviewed by us, and changes were made as stronger patterns linking codes to themes emerged from the data. Both researchers have a background in physiotherapy and have published on the quality use of pain medicines.¹⁷⁻¹⁹ Data saturation was reached (i.e. no new themes emerged).

Reference:

1. Hennink M, Kaiser BN. Sample sizes for saturation in qualitative research: A systematic review of empirical tests. *Soc Sci Med.* 2022 Jan;292:114523.

5. Why GPs with at least 5 patients in the last 12 months were considered? There is a rational for the number 5?

Response to reviewer: Our intention was to avoid recruiting GPs with no experience in managing low back pain – which arguably is next to impossible given the prevalence of back pain consultations in Australian general practice and the prevalence of the condition – and five is an arbitrary number. We have made that clearer in the manuscript.

We recruited 25 GPs registered to practise in Australia who reported having provided care for at least five patients with LBP with or without sciatica in the last 12 months. Four or less is what our team thought would be representative of little to no experience in managing LBP. We defined sciatica as a back problem with radiating pain below the knee with or without neurological symptoms (e.g. reduced muscle strength, sensation, or reflexes).¹²

6. Participant characteristics reported in Table 2 should be better described. For instance how Remoteness: Metropolitan/Rural was defined? how much is the range (minimum - maximum) for age, years of experience, workload? In the caption of Table 2 should be also reported that n=25 so that the Table could stand alone.

Response to reviewer: Metropolitan areas are defined following the recommendations of the Australian Statistical Geography Standard and encompass GPs working in Greater Capital City Statistical Areas. We have added that information to Table 2.

We have also added range (min-max) of experience and workload.

7. In Table 3 please report also % and not only counts.

Response to reviewer: We have provided counts in the table, in brackets. We have added “%” to each value to improve clarity.

8. In Table 3 is reported that in Vignette 3 for one GP it was not possible to record the interview for technical issues. The authors should report this in the methods part and better explain what's happened and why was not possible to record it again.

Response to reviewer: We provided more information in the first paragraph of the results. It reads as follows:

We interviewed 25 GPs from April to June, 2022. Most were males (64%) practising in metropolitan areas across six states of Australia. Their mean (SD) age was 53.4 (9.1), and they had a mean of 24.5 (9.3) years of practice as GPs. All but three worked full-time in private practices (Table 2). The recoding of one interview was mistakenly interrupted just before vignette #3 started and therefore there are only data from 24 GPs for that vignette.

9. In the discussion is not clear what are the novelties of this study and what this study in particular add to the literature, as main of the themes are already known in the literature and as the vignettes are also used in other studies.

Response to reviewers: Our findings align with some of the existing evidence around decision making in general practice; however, we believe that they provide valuable insights that could inform future work in the field of low back pain, where similar research was lacking prior to our study. One novel finding of our study, and one that we discussed in the manuscript, is the fact that many GPs decided which pharmacological treatment to prescribed based on a presumed pathoanatomical diagnosis for LBP. Our findings also help us understand what motivates GPs to prescribe certain medicines, which can inform the design of future interventions aimed at altering prescribing behaviours to better match new evidence and clinical practice guidelines.

We would also like to clarify that the vignettes used in our study were created by our study team, as described in the methods, for this study specifically. They have not been used before in other studies.

10. It could be interesting to discuss if prescriptions to real patients would differ from prescriptions using the vignettes.

Response to reviewers: We agree with the reviewer and have added the following to the discussion: However, the decision-making processes described in our study were anchored to the information provided in the vignettes and may not necessarily be generalisable to all LBP cases or truly reflect objective prescription data.³⁵ For example, while our study did show that GPs would most commonly prescribe or conditionally prescribe opioids for an acute exacerbation of chronic LBP (vignette 1) or subacute sciatica (vignette 2), which agrees with Australian data on prescription medicines for LBP,³ our finding that GPs would more commonly prescribe antidepressants for the patient with chronic LBP in vignette 3 differs from existing data. It is unknown whether this difference may be due to the characteristics of the vignette and reflective of practice for patients with the characteristics described in that vignette. Our sample was composed mostly of very experienced GPs practising in metropolitan areas. Views of more experienced GPs may differ from those in earlier stages of their careers.

11. Abstract line 29 - please correct "describe" with "described".

Response to reviewers: Corrected, thank you.

12. Line 41. Please check the sentence - I think "in" should be removed.

Response to reviewers: Corrected, thank you.

13. line 8 introduction I think that "or" should be deleted. Please check/rephrase the sentence

Response to reviewers: We have rephrased the sentence. It now reads as follows:

Australian data shows that 2 in every 3 patients who seek primary care are prescribed or recommended at least one pain medicine.²

REVIEWER 2

14. Thank you for this insight into general practitioner views on prescribing for lower back pain. The sample size is adequate for analysis for a qualitative study, and the authors have noted that men were overrepresented, and that rural GPs were underrepresented.

Response to reviewer: Thank you for your positive assessment of our manuscript.

15. The vignettes are realistic - in that they have tried other therapies prior to attending the GP. - these are all common scenarios.

Response to reviewer: Thank you. We piloted the vignettes with clinicians experienced in managing back pain to ensure that they were representative of common cases seen in daily practice.

16. Description of the themes emerging was executed well.

Response to reviewer: Thank you.

17. In the quotes, I can see no mention of heat packs, or referral to physical therapies -if these are not mentioned at all it would be worth mentioning specifically that they were not mentioned.

Response to reviewer: heat packs were not mentioned, but a small number of GPs did mention referral to physical therapies. We did not include that information as it is outside the scope of the manuscript, which focused on prescribing of medicines.

18. Did any specifically mention keeping active/ moving? If referral to rheumatologist, radiologist for steroid injection and neurosurgeon not mentioned also need to specifically note this (these patients would not necessarily require this).

Response to reviewer: Some of these were mentioned by a small number of GPs and did not emerge as either a theme or subtheme during data analysis.

19. There is one quote where the GP mentioned organising care for elderly mother. Although the focus is on medications these are all important aspects of therapy.

Response to reviewer: We agree with the reviewer. This quote was illustrative to theme 3, subtheme 2, 'psychosocial-based prescribing', highlighting that some GPs considered patients' social circumstances when deciding whether to prescribe a pain medicine and what to prescribe. It was interesting to note the range of psychosocial factors pointed out by GPs and their presence shaped GP's decision-making in different ways.

20. While GPs did not mention guidelines for treatment of back pain, most GPs would most likely have undertaken training with NPS on use of opioids and treatment of neuropathic pain -did any mention this? If so, this needs to be mentioned as was a large national program. NPS guidelines on lower back pain in primary care were released in 2011 but new framework was only released late in 2022 so GPs would not have seen this at the time of the study.

Response to reviewer: NPS Medicinewise training on use of opioids or management of neuropathic pain was not mentioned specifically by any GP. A very small number of GPs mentioned the Therapeutic Guidelines, but there was no other mention of other guidelines or specific training modules.

We have added some context to the results:

Subtheme 2 - Guidelines are unhelpful: Many GPs reported that guidelines were not helpful to their practice. Guideline recommendations were noted to be particularly unhelpful when patients required more than minimal intervention such as advice and simple analgesics. Recommended interventions such as advice and education about the benign nature of most LBP and advice to stay active were seen as problematic as it could be hard to convince patients in pain of their usefulness. One GP noted that the problem with guidelines was not necessarily their recommendations, but the time required to implement them in a context of GPs pressured for time. Clinical practice guidelines for LBP were not mentioned. Only the Therapeutic Guidelines were mentioned by a limited number of GPs.

21. Many of these GPs prescribed benzodiazepines - did any participants prescribe melatonin or other less addictive medications for insomnia? If so, this needs to be mentioned - also mention if they did (or did not) mention advice on sleep hygiene.

Response to reviewer: No other pharmacological treatment was mentioned for insomnia. Advice on sleep hygiene was not mentioned. We have added information on sleep hygiene to the results:

Subtheme 2 - Sleep patterns: Sleep influenced decision-making across the three vignettes, even though the patient in the chronic LBP vignette had no night pain or sleep disturbance. Improving sleep was an important treatment goal in the acute exacerbation vignette. For some, improving sleep was seen as the primary goal of treatment, and often led GPs to prescribe an opioid analgesics or antidepressants, or both. Advice on sleep hygiene was not mentioned.

22. In terms of prescribing and Subtheme 1, did any participants mention opiate contracts, practice policies or use of real time prescription monitoring (which at that time would have commenced in Vic and maybe in Qld but perhaps not yet in NSW).

Response to reviewer: None of those were mentioned by GPs. We note that we did have GPs from Victoria and Queensland in our sample.

23. Did any mention receiving notification regarding their opiate prescribing practices from the Commonwealth government as this program would have occurred prior to these interviews being undertaken.

Response to reviewer: Subtheme 4, "pressure from regulators" touches on that point. Some GPs expressed concerns about being pressured by regulators to not prescribe pain medicines such as opioid analgesics. One of the illustrative quotes that represent this subtheme mentions the 'Department of Health', but no specific programs rolled out by the Federal Government.

24. I subtheme 2 - did any mention any specific side effects, such as constipation, sedation (and need for care with driving), addiction or other side effects - If not, need to also mention this.

Response to reviewer: Yes those were frequently mentioned. We have now mentioned them in the results – see below. We note that subtheme 2 describes how some GPs acted based upon their knowledge about side effects, particularly in relation to opioids.

Subtheme 2 - Side effects of prescription pain medicines: Similarly, most concerns about side effects were related to opioid analgesics. Many GPs expressed concerns about the safety profile of specific opioid analgesics, most commonly codeine, and made decisions about which opioid analgesic to use based on their knowledge about the safety profile of opioid analgesics (e.g., sedation, constipation, nausea). In contrast, very few GPs expressed concerns about other pain medicines with potential for misuse and abuse, such as pregabalin.

25. Subtheme - stigma about using antidepressants - the mention of suggesting a mental health diagnosis is raised - did any GPs mention other flowon effects such as effect on the person's life insurance (increases cost of insurance if existing mental health condition).

Response to reviewer: Those were not mentioned by any GP.

26. Did any GPs mention risk to themselves or that there is a risk of verbal abuse or risk of physical harm with some clients if medications not prescribed (not a major point but definitely occurs).

Response to reviewer: Those were not mentioned by any GP.

27. In particular, the paragraph regarding using scans to guide treatment is relevant and discussed valid on this point.

Response to reviewer: Thank you.

28. The paper would have benefited from a general practitioner input, to interpret the findings so as to understand the context of general practice. Best practice in research involving primary care is to include practitioners on the research team.

Response to reviewer: We agree with the reviewer. Since this study was conceived in 2021, we have changed our approach and have been collaborating with general practitioners in projects done in general practice or involving general practitioners.

VERSION 2 – REVIEW

REVIEWER	Di Gangi, Stefania University of Zurich
REVIEW RETURNED	23-Aug-2023

GENERAL COMMENTS	Authors have mostly addressed the Reviewers' comments. 1) However, they forgot, in Table 3, to report % in addition to count. In fact the title of The table "Frequency" is not so appropriate. So please add also % (as authors did in Table 2) 2) Line 55, pag 4, please add patients otherwise is not clear " Four or less patients is what our team .." After these corrections, the manuscript would be suitable for publication.
---

VERSION 2 – AUTHOR RESPONSE

REVIEWER 1

1. However, they forgot, in Table 3, to report % in addition to count. In fact the title of The table "Frequency" is not so appropriate. So please add also % (as authors did in Table 2)

Response to reviewer: We have amended the title of table 3 and have added the percentages in all cells.

2. Line 55, pag 4, please add patients otherwise is not clear " Four or less patients is what our team .."

Response to reviewer: We have added "patient" to the sentence, which now reads as follows: "Four or less patients is what our team thought would be representative of little to no experience in managing LBP."